# Pulmonary Nodule Detection in CT Images using Dual Path U-Net and Multiscale Region Proposal Network

**Jiahua Xu**[*1]                                                                JIAHUA.XU@OVGU.DE
**Philipp Ernst**[*1]                                                          PHILIPP.ERNST@OVGU.DE
**Eric Liu**[*1]                                                            TUNG-LUNG.LIU@ST.OVGU.DE
**Andreas Nürnberger**[1]                                      ANDREAS.NUERNBERGER@OVGU.DE
[1] *Faculty of Computer Science, Otto von Guericke University Magdeburg, Germany*

**Editors:** Under Review for MIDL 2021

## Abstract

Pulmonary cancer is one of the most commonly diagnosed and deadly cancers and often diagnosed by incidental findings with computed tomography. Automated pulmonary nodule detection is an essential part of computer-aided diagnosis, which is still facing great challenges and difficulties to quickly and accurately locate the exact nodules' positions. This paper proposes a novel deep learning model based on a Dual Path network in a U-Net structure generating multiscale feature maps as well as taking advantage of having 2.5D input to provide better contextual information. An extended upsampling strategy is proposed to minimize the ratio of false positives and maximize the sensitivity for lesion detection of nodules. The results show that our new upsampling strategy improves the performance by having 85.3% sensitivity at 4 FROC per image compared to 84.2% for the regular upsampling strategy as well as 81.2% for VGG16 based Faster-R-CNN.

**Keywords:** Pulmonary nodule detection, Dual Path U-Net, Region Proposal Network.

## 1. Introduction

Pulmonary cancer is one of the most commonly diagnosed and deadly cancers among other cancers in medical research (Ferlay et al., 2015). Other pulmonary diseases such as COVID-19 (Lu Wang et al., 2020) or pulmonary infection may also cause serious damage to the lung. The most common problem during diagnostic in radiology is solitary pulmonary nodules, i.e. single, round or oval nodules generally smaller than 3 cm (Toghiani et al., 2015).

Computed tomography (CT) is one of the most common non-invasive screening approaches for diagnosing pulmonary diseases (Midthun, 2016). Pulmonary nodules that appear on the images have a high variability in terms of size, shape and location in the pulmonary regions (Rubin, 2015). Radiologists need to go through these series of images and make annotations on suspicious lesions. This process requires radiologists to carefully observe the images leading to a high workload, while annotating depends on individual radiologists' experiences and might lead to a high possibility of error, especially for small nodules (approx. 5 mm) (Al Mohammad et al., 2017).

The number of patients diagnosed with pulmonary nodules increased as more patients with a high risk of lung cancer have low-dose CT scans acquired for the screening of cancer (Yuan et al., 2017). Computer-aided diagnosis (CAD) systems support radiologists at both

---

[*] Contributed equally

nodule detection and localization tasks. However, it is challenging to reduce the misdiagnosis and false positives (FPs) in early-stage lung cancer diagnosis (Dou et al., 2017). The main issue which leads to a high ratio of FPs, particularly for pulmonary nodule detection, comes from the variability of nodules in terms of size, shape and location (Rubin, 2015) and, compared to regular RGB images, gray-scale medical images provide less information in terms of edges and textures to distinguish different tissues (Escobar, 2008).

In recent researches, Convolutional Neural Networks (CNNs) have greatly improved performance and efficiency in CAD systems (Zhao et al., 2019). Variations based on CNNs (Xie et al., 2019), U-Net (Zhao et al., 2018) or Faster R-CNN (Ding et al., 2017) have been reported. Other studies also show that the performance of nodule detection and classification on medical images using 3D-CNN models, which can provide more information for the model, is generally better than the 2D-CNN model, such as in (Gu et al., 2018; Nasrullah et al., 2019; Setio et al., 2017; Huang et al., 2017). However, 3D-CNN or 3D-Dual Path Network (Zhu et al., 2018; Jiang et al., 2020) models require more computational resources and memory than 2D-CNN models due to a larger number of network parameters (Kamnitsas et al., 2017). In contrast, a compensation approach of 3D-CNN and 2D-CNN models makes use of 2.5D-CNN models that can take advantage of inter-slice features compared to 2D-CNN models and have lower memory requirements than a 3D-CNN model.

In this paper, we adopt ideas from different algorithms based on deep learning and propose a network using Faster R-CNN (Ren et al., 2017) with Dual Path Network (DPN) (Chen et al., 2017) as the backbone in a U-Net (Ronneberger et al., 2015) structure while having 2.5D medical images as input and multiscale feature maps. The main idea is to utilize the volumetric and contextual information around the nodules as well as from multiple views of CT scans in order to reduce the FP rate and increase the true positive (TP) rate of detected pulmonary nodules while keeping the computational effort of the method low.

## 2. Method

### 2.1. Dataset: DeepLesion

DeepLesion (Yan et al., 2018b) was released by the National Institutes of Health Clinical Center. It consists of 32,120 axial CT slices from 10,594 CT scans (studies) from 4,427 unique patients. There are 1–3 lesions such as lung nodules, liver tumors or enlarged lymph nodes in each image with accompanying bounding box coordinates and size measurements, adding up to 32,735 lesions altogether. We have extracted CT images that are annotated with pulmonary nodules, which resulted in 2,394 CT images for our dataset in our case. 1,916 CT images are used for training and 478 CT images for validation.

### 2.2. Data Preprocessing

The CT slices of DeepLesion are normalized by converting Hounsfield units to mass attenuation coefficients and dividing by the 99th percentile of the entire dataset. Each slice has 1 mm to 5 mm thickness in most cases while some of the images are 0.625 mm or 2 mm. For each lesion, there is one key slice with 30 mm of extra slices in front of and behind the key slice. However, only the key slice has the annotation data including lesion types, coordinates of 2D bounding-boxes and RECIST diameters for the lesions. The CT slices

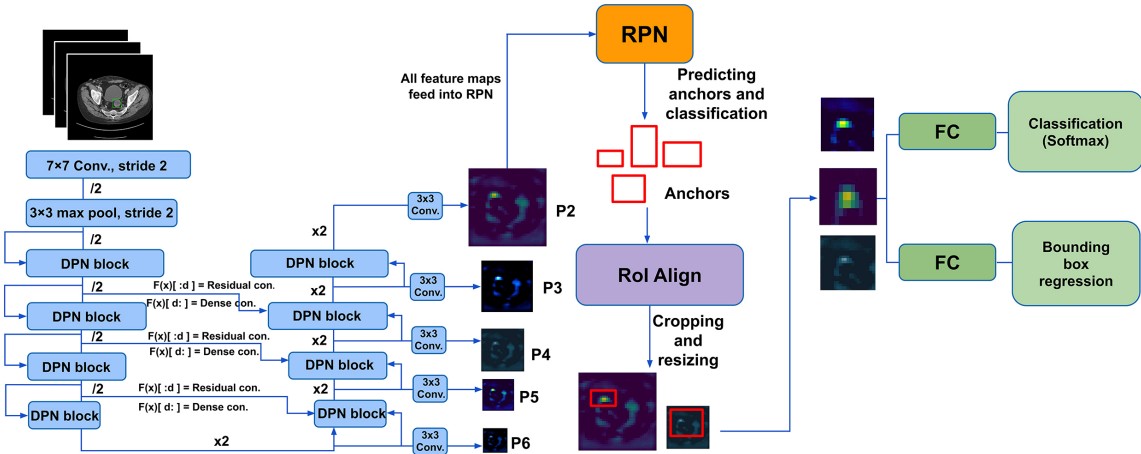

Figure 1: Our model generates five feature maps for RPN. ROI Align crops the feature maps according to the sizes of the anchors on the corresponding feature map.

were resized to $512\,\text{px} \times 512\,\text{px}$. For training, we adapted data augmentation where images are flipped horizontally and vertically and rotated with a probability of 50% respectively to enrich the variability of the CT images. During augmentation, the corresponding coordinates of the bounding boxes are updated as well. To enhance the spatial information, we concatenate one more slice in front of and behind the key slice to get a 2.5D model. This approach increases the information for the model while being a lot more lightweight than a 3D model.

## 2.3. Proposed Model

We adopt ideas from different algorithms to tackle the issue of variability of pulmonary nodules. The two-stage object detection model Faster R-CNN is used, which is hoped to reduce the ratio of FPs by providing more spatial and contextual information. Fig. 1 shows the outline of the architecture of the proposed model.

The backbone architecture of the first stage is a combination of U-Net and DPN with skip connections between the encoding and decoding path to provide more spatial information and to improve the flow of gradients. Five feature maps of different resolutions are derived during the decoding process that are fed into a Region Proposal Network (RPN) (Ren et al., 2017) for the first part of classification and anchor box regression.

ROI Align from Mask R-CNN (He et al., 2017) crops and resizes the regions of interest from RPN to a fixed resolution. Eventually, two sets of fully connected layers, which classify nodule or non-nodule, are attached at the end of our network for the second part of classification and anchor box regression, respectively.

### 2.3.1. Encoder

The encoder is a modified DPN architecture that performs DenseNet and ResNet in parallel. The model starts with having 64 kernels with a 7x7 convolutional layer and a stride of 2 followed by a 3x3 max pooling layer with a stride of 2 subsequently as an initial block. Afterwards, there are four more bottleneck blocks (Chen et al., 2017) having 1x1, 3x3 and 1x1 convolutional layers where each convolutional layer is followed by batch normalization and ReLU activation. Grouped convolutions on all channels are performed within the 3x3 convolutional layer like in ResNet (Xie et al., 2017).

### 2.3.2. Decoder

The decoder stages consist of scaling up the feature maps and concatenating or adding the skip connections in the same encoder stage, where the upsampling starts after scaling up the feature map, as shown in Fig. 2 (Part A, Type I). This approach is relatively straightforward and easy to implement since ResNet-like, DenseNet-like or FCNs have only a single type of operation to combine the different connections, yet, DPN has both of these operations during encoding. Furthermore, there is no particular guideline indicating how to connect every stage in a U-Net architecture having DPN as a backbone for decoding. The implementation of starting the shortcut before the upsampling layer is shown in Fig. 2 (Part B, Type II). Hence, it provides us a discussion space to observe the performance if we start the shortcut before upsampling and implement both addition and concatenation to connect with the skip connection as a regular DPN block during decoding. Finally, an extra 3x3 convolutional layer is attached after concatenating or adding the skip connections.

We have implemented two different types of upsampling with DPN blocks as the backbone to observe the performance on pulmonary nodule detection. Type I is a regular upsampling approach and starts the DPN block after upsampling as shown in Fig. 2 (A) on the left. For Type II, we extract a part of the data stream before upsampling, where the main data stream is upscaled in resolution by a DPN block. This is done by the 3x3 convolutional layer within the bottleneck in a DPN block with a stride of 2. The detailed visualization of our proposed upsampling strategy is depicted in Fig. 2 (B), which shows how the two operations of concatenation and addition interact during encoding and decoding.

Appx. A explains in more detail how our DPN architecture differs from the original to be less computationally expensive.

### 2.4. Generating Ground Truth Labels

For training, generating ground truth labels for a single scale feature map in an FPN architecture is relatively easier than in a multiscale architecture because the mechanisms in these architectures are fundamentally different. For a single scale architecture, all labels for classification or regression can be concatenated. However, this approach is impossible for a multiscale architecture owing to the fact that each scale of an anchor with three different ratios is performed on one single level of the feature map. The concatenation for training labels is not applicable because the height and width are different on each level.

For this reason, we vectorized all labels and performed the concatenation on the only dimension. However, this approach requires a lot more attention to making sure the order

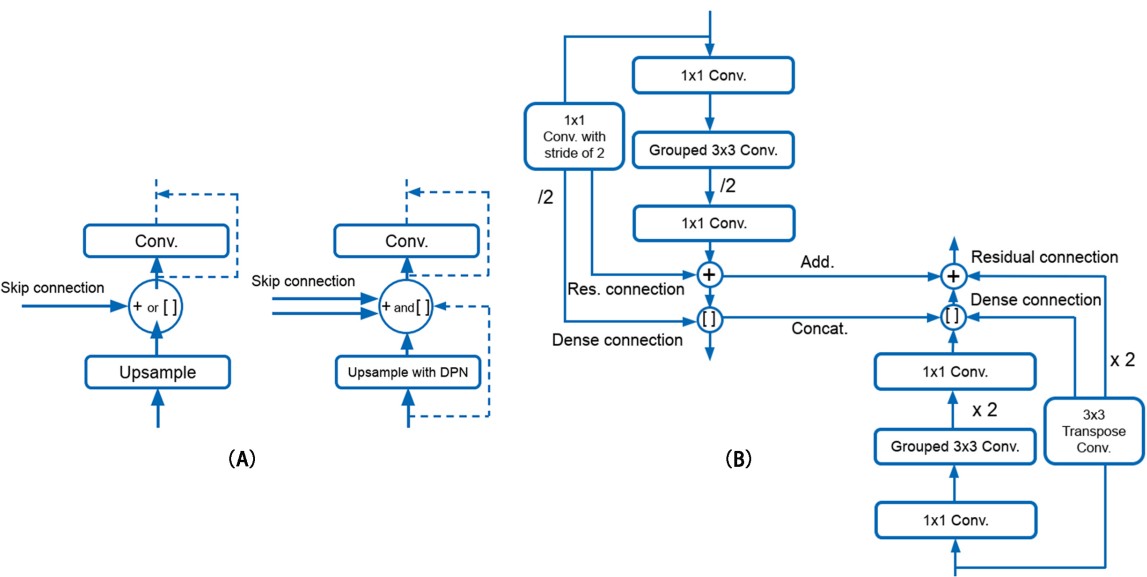

Figure 2: (A) Upsampling approaches: scaling up the feature map and short-cutting afterwards (Type I, left), short-cutting before upsampling with both addition and concatenation (Type II, right). (B) Encoding (left) and decoding process (right) on the same network level.

of the labels is valid in the order of how convolution filters are doing windowing on feature maps from different levels.

For RPN, we mark an anchor as positive if the IoU is more than 0.7 with the ground truth bounding box and as negative if the IoU is lower than 0.3. Those anchors having IoU between these maximum and minimum threshold are considered as neutral, and are not involved for training. In addition, the regression parameters are only calculated for the positive anchors because, by nature, there are a lot more negative samples than positive ones, and we did not want our model to learn too much about the negative samples. Determining the optimal sizes and ratios of the anchors is discussed in Appx. B.

The sub-network Classifier relies on the prediction of RPN to provide information on the location of the target. We considered locations as background if the IoU was between 0.3 and 0.5 and foreground if the IoU was greater than 0.5. Only positive samples calculate the regression parameters in this stage as well.

**2.5. Loss Function**

Our loss function follows the definition of the multi-task loss function of Faster R-CNN (Ren et al., 2017):

$$L(p_i, t_i) = \frac{1}{N_{cls}} \sum_i L_{cls}(p_i, p_i^*) + \lambda \frac{1}{N_{reg}} \sum_i p_i^* L_{reg}(t_i, t_i^*)$$

It consists of the two parts $L_{cls}$ (for RPN and Classifier) and $L_{reg}$ (for RPN), where $i$ is the index of an anchor in a mini-batch and $p_i$ is the predicted probability of anchor $i$. The ground truth label $p_i^*$ is 1 if the anchor is a positive sample and 0 if negative. $t_i$ is the four coordinates of the predicted bounding box and $t_i^*$ is that of the ground truth box associated with a positive anchor.

The classification loss $L_{cls}$ is a log loss over two classes (object vs. not object) for RPN and multiple classes for Classifier. For the regression loss $L_{reg}$, we used the smooth L1 loss (Girshick, 2015). The term $p_i^* L_{reg}$ calculates the regression loss only for positive anchors ($p_i^* = 1$). $\lambda$ is set to 1 since it was proven insensitive in (Ren et al., 2017).

For bounding box regression, we adopted the regression parameters of the 4 coordinates following (Girshick et al., 2014), where $x$, $y$, $w$, and $h$ denote the box's center coordinates and its width and height. The variables $x$, $x_a$, and $x^*$ denote the predicted box, anchor box and ground truth box respectively (same for $y$). The parameters are calculated as:

$$t_x = (x - x_a)/w_a \quad t_y = (y - y_a)/h_a \quad t_w = \log(w/w_a) \quad t_h = \log(h/h_a)$$
$$t_x^* = (x^* - x_a)/w_a \quad t_y^* = (y^* - y_a)/h_a \quad t_w^* = \log(w^*/w_a) \quad t_h^* = \log(h^*/h_a)$$

### 2.6. Metric

The free-response receiver operating characteristic (FROC) is one of the standard metrics in lesion detection (Setio et al., 2017). Its evaluation is performed by measuring the sensitivities (%) with respect to their corresponding average FP rate per scan. TPs and FPs are determined by thresholding a confidence measure of the predictions. For our evaluations, we calculate the IoU of the predicted bounding boxes with the ground truth bounding boxes. If it is larger than 0.5, it represents a TP and an FP otherwise.

### 2.7. Experimental Setup

For training, we use a weight decay of $1 \times 10^{-4}$. The initial learning rate $1 \times 10^{-3}$ is reduced to $1 \times 10^{-4}$ after 5 epochs. Due to time limitations, all models were trained with 15 epochs on the training dataset. We use Adam as optimizer as well as data augmentation, dropout and normalization to prevent overfitting. All trainings and tests were performed on Google Colab Pro (NVIDIA Tesla K80) utilizing Keras 2.3.1 with tensorflow as backend.

### 3. Results

We implemented three different models as well as state-of-the-art 3D models to compare the performance. Our baseline model is based on Faster R-CNN with VGG16 as backbone where the prediction was only performed on the last feature map. The other two models are our proposed models with the regular upsampling strategy (Type I) and the proposed upsampling approach (Type II).

Tab. 1 shows the the sensitivities at 1/2, 1, 2, 4, 8 and 16 average FPs per scan to compare our models and state-of-the-art 3D models. Note that DeepLesion is a universal lesion dataset. The results from other researches are based on multiple types of lesions, while Tab. 2 shows the comparison only for pulmonary nodules at 4 average FPs per scan.

The first row of Fig. 3 visualizes the detection results in the official test dataset of DeepLesion for the baseline model while the second row of Fig. 3 shows the results of the

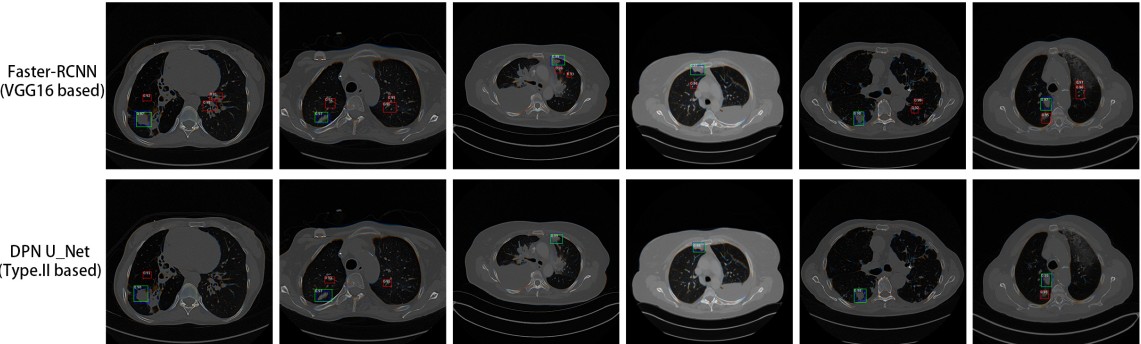

Figure 3: Exemplary predictions on the CT slices using VGG16 based Faster-RCNN and DPN U-Net (Type II).

Table 1: Sensitivities at different FPs per image (multiple lesions).

| Model | Sensitivity (%) at FPs | | | | | |
|---|---|---|---|---|---|---|
| | 0.5 | 1 | 2 | 4 | 8 | 16 |
| Original R-FCN (Yan et al., 2018a) | 55.7 | 67.2 | 75.3 | 82.2 | 86.2 | 89.1 |
| 3DCE (Yan et al., 2018a) | 62.4 | 73.3 | 80.7 | 85.6 | 89.0 | 91.0 |
| 3DCE_CS_Att (Tao et al., 2019) | 71.4 | 78.5 | 84.0 | 87.6 | 90.2 | 91.4 |
| **VGG16 (baseline)** | 55.8 | 66.3 | 74.7 | 81.2 | 84.8 | 87.5 |
| **DPN U-Net, Type I** | 60.9 | 71.3 | 77.7 | 84.2 | 87.6 | 88.9 |
| **DPN U-Net, Type II** | 64.6 | 74.1 | 80.7 | 85.3 | 88.3 | 89.8 |

proposed model DPN U-Net Type II. Blue, green and red boxes represent the ground truth, TP and FP boxes respectively, and the number on the top-left corner of the boxes represents the confidence. We also show some intermediate feature maps in Appx. C for DPN U-Net Type II.

## 4. Discussion

From Tab. 1, we see that the average FROC of DPN U-Net Type II yields 80.5%, surpassing the DPN U-Net Type I with 78.4%, and the baseline model with 75.1%. The baseline model is a single scale model where the RPN only looks for targets on the same resolution of the feature map. By looking at Fig. 3, we can see that the baseline model generates more FP predictions than DPN U-Net Type II on the same CT images in general. Our results show that multiscale feature maps can help to improve the performance.

Our best result of the proposed model is not surpassing the best result of the 3D model. 3DCE_CS_Att with 21 slices for pulmonary nodule detection achieves a sensitivity of 92% at 4 FPs while having average FROC of 83.9% for all lesions from DeepLesion. Yet, the sensitivity of DPN U-Net Type II at 4 FPs is at 85.3%, which is already quite close to 3DCE with 89%. By theory, 3D input provides more contextual information for a deep learning

Table 2: Sensitivities at 4 FPs per image (pulmonary nodules only).

| Model | Sensitivity (%) at 4 FPs |
|---|---|
| 3DCE (Yan et al., 2018a) | 89.0 |
| 3DCE_CS_Att (Tao et al., 2019) | 92.0 |
| **DPN U-Net Type II (Ours)** | 85.3 |

model, yet, it also requires more computational resources. We assume that our model might approach the performance of the 3D network closer if we make it deeper or wider.

The traditional upsampling approach in Type I might lose some information when upsampling, although it has the skip-connection in the same stage. DPN U-Net Type II intends further to provide more contextual information upon the original structure. The results show that Type II is an efficient approach to reuse the DPN block and can provide more contextual information by adding a shortcut connection before upsampling. Furthermore, during the experiment, DPN U-Net Type I and II require a similar computational time per batch, while the baseline model requires only 50% of the time for computation. This behavior is expected since DPN consists of more complex operations and structure.

## 5. Conclusion

This paper proposed a DPN U-Net which takes advantage of multiscale feature maps to locate pulmonary nodules in various shapes and sizes. Our work shows that the proposed model DPN U-Net Type II surpasses the results performed by the single-scale feature map model. Our new approach for upsampling improves the performance of the traditional upsampling strategy. The proposed model DPN U-Net Type II reuses the DPN block throughout the whole network, which is an efficient way to explore new potential features and prevent vanishing gradients by having both operations from ResNet and DenseNet. 2.5D input is an adequate compensation in terms of contextual information and computational resources. Although our proposed model DPN U-Net Type II, which yields the best result at 80.5% in our work, is not surpassing the performance of state-of-the-art 3D-models, our results show that it still has the potential to be developed further. The sensitivity at 4 FPs of DPN U-Net Type II is at 85.3%, which is already quite close to 3DCE with 89%. Overall, we assume that the performance of our proposed model might still be improved if we make the model deeper or wider and train it in more epochs.

## Acknowledgments

This work was conducted within the International Graduate School MEMoRIAL at OVGU Magdeburg, supported by the ESF (project no. ZS/2016/08/80646).

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

## Appendix A. Architecture Optimization

The original DPN architecture with 92 layers yields the best results in the authors' experiments, which is both a deep and wide network. To reduce the computational effort, we

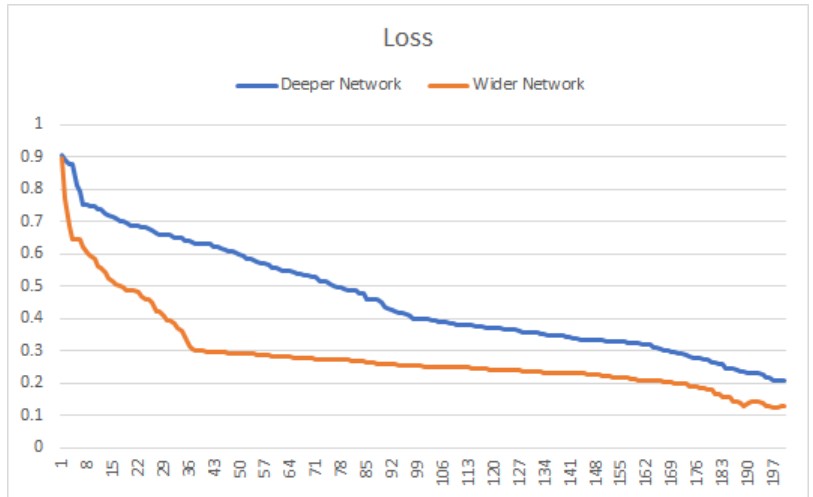

Figure 4: Comparison of losses for two different implementations

| 7 x 7, 64, stride 2 |
| 3×3max pool, stride 2 |

$$\begin{bmatrix} 1 \times 1, 24, \\ 3 \times 3, 24 \ \ G=8 \\ 1 \times 1, 32 \ (+8) \end{bmatrix} \times 3$$

$$\begin{bmatrix} 1 \times 1, 48 \\ 3 \times 3, 48 \ \ G=8 \\ 1 \times 1, 64 \ (+16) \end{bmatrix} \times 4$$

$$\begin{bmatrix} 1 \times 1, 96 \\ 3 \times 3, 96 \ G=8 \\ 1 \times 1, 128 \ (+32) \end{bmatrix} \times 5$$

$$\begin{bmatrix} 1 \times 1, 192 \\ 3 \times 3, 192 \ G=8 \\ 1 \times 1, 256 \ (+64) \end{bmatrix} \times 3$$

Figure 5: Architecture of our DPN blocks

evaluated how depth and width influence the performance and modified our architecture to be either deep or wide.

For a wider network, we set the repetitions of the bottleneck in the stages to [4, 4, 4, 4] and base filter numbers and filter increments to [32(+8), 64(+8), 128(+8), 256(+8)]. For a deeper network, the repetitions were [8, 8, 8, 8] with base filter numbers and filter increments of [16(+8), 32(+8), 64(+8), 128(+8)].

The models were trained on a small subset of 200 CT slices for training and 20 CT slices for testing. We used the same loss function as in Faster R-CNN. Fig. 4 shows the comparison of losses for the two different strategies.

The wider model achieves an accuracy 60.3%, opposed to 55.8% for the deeper model on the test dataset. For this reason, we focused on the width of our network and set the architecture of our DPN blocks to be like in Fig. 5.

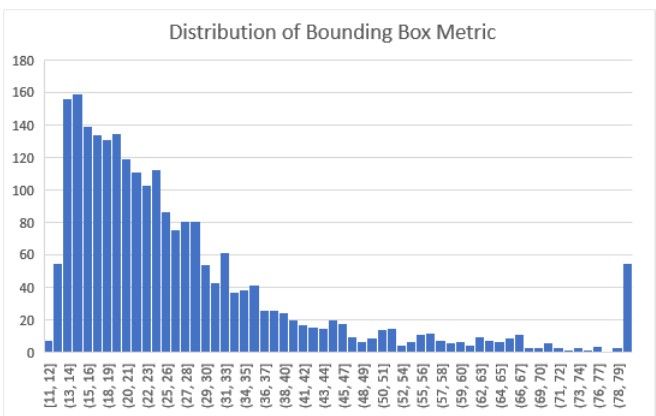

Figure 6: Distribution of the bounding box metric for ground truth boxes.

## Appendix B. Anchor Set Generation

Deciding on the sizes of anchors plays a crucial role in an object detection architecture, especially in an FPN architecture since there is only one anchor scale with three different ratios performed on a single resolution of feature maps from different levels. Furthermore, the entire network relies on the first sub-network RPN using anchors to locate the target. It provides information about the location of the target for the bounding box regression as well as for the second part of bounding box regression in the later sub-network Classifier.

A set of poorly designed anchors can lead to very long trainings and affect classification performance as well. In addition, the anchor sizes are highly dependent on the target sizes in the training dataset. In other words, anchors which are used in other object detection models might not be suitable for our data.

In the DeepLesion dataset, ground truth bounding box coordinates are provided as sequences of $(x_1, y_1, x_2, y_2)$ in pixel scale according to the original size of CT slices $(512 \times 512)$ for all pulmonary nodules. A simple bounding box metric (BBM) was calculated as:

$$BBM = \sqrt{(x_2 - x_1) \cdot (y_2 - y_1)}$$

The distribution of this metric is shown in Fig. 6, where most of the BBMs are between 10 px and 40 px. A grid search was performed on different sets of anchor scales to find an optimal set. We define positive anchors as those having an IoU above an upper threshold $(> 0.7)$ and negative anchors as those below a lower threshold $(< 0.3)$. We also compare with the anchor set $[32, 64, 128, 256, 512]$ that is used in the original paper of Mask R-CNN. Tab. 3 shows some results this experiment. The set $[50, 65, 100, 150, 190]$ yields the best result for average positive anchors. Therefore, it is our choice for anchor generation. Like in (Ding et al., 2017), we use anchor ratios of $[0.5, 1, 2]$ for each anchor scale.

Fig. 7 (a) shows an example of a CT slice, where the red rectangle indicates the ground truth bounding box. Fig. 7 (b) shows the generated anchor boxes on the CT slice and Fig 7 (c) shows the anchor box regression during training getting closer to the ground truth bounding box. Dotted lines indicate the learning process with the yellow box in the center being the ground truth bounding box.

Table 3: Average anchors in different scales

| Anchor scale | Avg. positive anchors | Avg. negative anchors |
|---|---|---|
| $[50, 65, 100, 150, 190]$ | **98.7** | 157.2 |
| $[32, 64, 128, 256, 512]$ | 31.3 | 224.6 |
| $[10, 20, 50, 60, 70]$ | 2.36 | 253.5 |

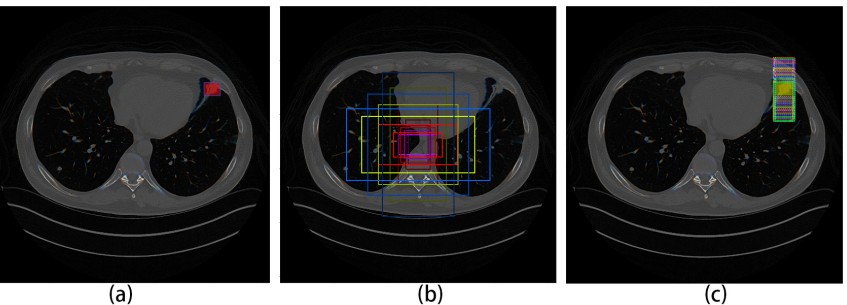

Figure 7: Visualization of bounding boxes and anchors: (a) ground truth bounding box (b) generated anchors (c) anchor regression during training.

## Appendix C. Visualization of Feature Maps

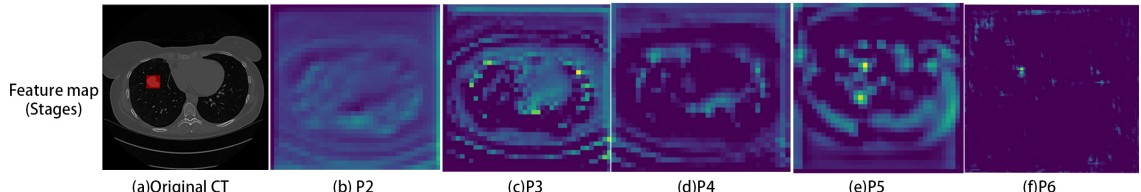

Figure 8: P2, P3, P4, P5 and P6 feature maps from DPN U-Net Type II in different stages.

