# OpenReview forum: "Pulmonary Nodule Detection in CT Images using Dual Path U-Net and Multiscale Region Proposal Network"
_MIDL.io/2021/Conference — Submitted to MIDL 2021_

### Official Review · AnonReviewer1 · 2021-03-07

**Confidence:** 3
**Preliminary Rating:** 2
**Final Rating:** 1

**Summary:**

This paper presents a method for pulmonary nodule detection in chest CT scans. The approach is based on the Faster R-CNN framework but uses a special backbone network that is similar to a U-net. The input to the network are 3 consecutive 2D slices. The approach is trained and evaluated with part of the DeepLesion, namely the approximately 2400 CT scans with pulmonary nodules.

**Strengths:**

* The method is clearly presented.
* The paper uses the DeepLesion dataset, which includes a large number of pulmonary nodules.
* Overall, the method achieves reasonable detection performance in comparison with other methods.

**Weaknesses:**

* The presented combination of U-net-like backbone and the Faster R-CNN detection framework does not outperform the baseline. The paper speculates that the approach is still promising and that a wider or longer trained network might perform better. It would have been more useful to first verify this and include these results in the paper, these experiments do not seem very complicated.
* The introduction mentions that the main idea of this approach is "to utilize the volumetric and contextual information around the nodules as well as from multiple views of CT scans". However, including slices before and after the key slice is something that was already done in the very first baseline for the DeepLesion dataset (Yan et al., 2018b)
* Since there are no slices without any lesions in the DeepLesion dataset, it is hard to compare the detection performance with other nodule detection systems.


**Deanonymize Review:**

no

**Detailed Comments:**

* The training/validation/test split is not really clear, the paper states that around 2400 CT slices with nodules were used, and almost 500 of them for validation. But it is not really clear if the final results were computed on the official test set or on the validation set.
* Figure 3 only shows examples of relatively large nodules while detecting small nodules is quite relevant, especially in lung cancer screening. It would also be useful to add the meaning of the colored boxes to the figure caption.


**Final Rating Justification:**

The paper proposes a lesion detection approach and the main point seems to be that a U-net-like backbone outperforms a VGG-like backbone in combination with Faster R-CNN. This is all trained/evaluated in 2D, which seems not very useful given that 3D models outperform 2D models. Also, contributions and results are not described well, it is hard to understand what point the paper tries to make (e.g., the title mentions pulmonary nodules but more results are presented for a mix of lesions of which pulmonary nodules are only a subset).

**Justification Of The Preliminary Rating:**

This paper presents a lesion detection approach that is quite similar to other approaches that were proposed for the same dataset. The paper only speculates that this method might outperform other approaches when trained longer or when using a wider network, but does actually not introduce a very innovative approach and additionally reports worse performance than similar prior approaches.

**Paper Type:**

methodological development

**Special Issue:**

no

---

> ### Author Response · Authors · 2021-03-16
> **Feedback reviewer 1**
>
> We appreciate your comment and would like to address your concerns as follows:
>
> Weakness 1. In fact, the proposed Type II network outperformed the baseline (VGG16-based Faster R-CNN). We assume that Table 1 and 2 might be prone to be misunderstood. Since our main contribution is to reduce the FP rate with the new upsampling method, we take two methods as our baseline: Faster R-CNN and Type I (Dual Path Unet with single connections). The results (In Table 1 and Figure 3) show that our proposed upsampling scheme way reduces the FP rate with higher accuracy. We conclude that this new upsampling could help improve future research. As discussed in the comments of the other reviewers, we did not state our method as the SOTA and discussed all the possibilities that could contribute later. Simply a combination of Dual Path and Unet is not our main highlight, we also noted some paper has done that before [1], though in a totally different domain. However, the results of Dual Path connections using our Type II upsampling proved that we could reduce the FP rate for the nodule detection.
>
> Weakness 2. We agree with you and also cited the paper you listed. We do not claim this idea to be our novelty. The idea of utilizing 2.5D input is not our primary goal in this paper either, considering the GPU and computation resources. We only try to keep the resources low by utilizing the volumetric and contextual information around the nodules as 2.5D input. Since all the models we proposed were evaluated with 2.5D input and the original DeepLesion model used the same approach, it allows us to directly compare these models.
>
> Weakness 3. We agree with your comment, but comparing with other nodule detection systems is not our primary target either, since there are no control slices. We mainly contribute the proposed upsampling strategy to reduce the false positive rate as shown in Figure 3, where the performance is much better than for the baseline, visually, and also quantitatively as shown in Table 1.
>
> Detailed comment 1. Thanks for pointing that out, we will add a more clear description in the data preprocessing part. We have extracted only those CT images that are annotated with pulmonary nodules which resulted in 2,394 CT images. We split 1,916 CT images for training and 478 CT images for validation.
>
> Detailed comment 2. We will add the meaning of the colored boxes to the caption as follows: Blue, green and red boxes represent the ground truth, true positive and false positive boxes respectively, and the number on the top-left corner of the boxes represents the confidence. We will also replace some of the exemplary slices with ones that contain smaller nodules.
>
>
> Justification of the preliminary rating:
> First of all, we believe it is our fault that we did not make our main contribution clear, and also listed the Table in an unreasonable way, we will correct this in the new version of the paper.
>
> ‘This paper presents a lesion detection approach that is quite similar to other approaches that were proposed for the same dataset.’
>
> Yes, several similar but definitely technically different works have been done and with higher accuracy, we also discuss this in the discussion part. We also stated that the result is not SOTA. However, in our paper, we have no initiative to compare with them but refer to them. Because our baseline is VGG16-based Faster-R-CNN and Type I network combining the Dualpath Unet, our final proposed network performed much better than the two baselines as shown in Table 1 and reduced the FP rate as depicted in Figure 3.
>
> ‘The paper only speculates that this method might outperform other approaches when trained longer or when using a wider network, but does actually not introduce a very innovative approach and additionally reports worse performance than similar prior approaches’.
>
> For future work, we discuss the possibility of potential approaches that could improve the overall accuracy. Since we were limited by our GPU at the time we performed the trainings, we could not evaluate more complex networks in our paper. We will do so later, since we now have more GPUs available. Table 1 and 2 are a bit misleading at the moment, which we are going to address in a new version of the paper.
>
> [1] Y. He, X. Yu, C. Liu, J. Zhang, K. Hu and H. C. Zhu, "A 3D Dual Path U-Net of Cancer Segmentation Based on MRI," 2018 IEEE 3rd International Conference on Image, Vision and Computing (ICIVC), Chongqing, China, 2018, pp. 268-272, doi: 10.1109/ICIVC.2018.8492781.

---

> > ### Comment · AnonReviewer1 · 2021-03-19
> > **Follow-up question**
> >
> > Thank you for your elaborate reply. If I understand correctly, you are trying to improve 2D slab approaches (you refer to them as 2.5D) but admit that you cannot reach the performance of 3D approaches with this strategy. Okay. However, what I find confusing is that you list results for your 2.5D baseline (VGG) only for all types of lesions in Table 1, but list results for pulmonary nodules that are the focus of the paper only for 3D networks and your proposed approach (Table 2). Why is the VGG baseline model missing from Table 2?

---

### Official Review · AnonReviewer3 · 2021-03-08

**Confidence:** 5
**Preliminary Rating:** 2
**Final Rating:** 2

**Summary:**

The authors tackle the issue of Pulmonary Nodule detection, a very relevant task in medical imaging with much competition.
They combine several well known methods in the field (F-RCNN, Dual path networks + 2.5 d inputs) to achieve good, although not near state-of-the-art, results while keeping computation times low.

**Strengths:**

The paper applies the state of the art, and finds an innovative combination of these methods and applies it to a relevant topic. This is a more application oriented paper, but can deliver very relevant information to the field on what combination of methods works well (this is highly non-trivial).
The results seem good, compared to certain comparable methods, although not compared to the best methods. And they compare their specific upsampling method to another baseline method using another upsampling, and compare favourably. This might be useful for the field to know.

**Weaknesses:**

The results are not as good as the 3D model (3DCE) as noted by the authors. In the discussion the authors hypothesise that their method might work better with more layers, but this is of course hypothetical.

The paper is very implementation focused, but some details seem glossed over. The experimental setup notes that augmentations and normalisations are used, and the Adam optimizer is used. However specifics are not clear -- like which augmentations, which normalisations, what parameters for Adam besides the learning rate?



**Deanonymize Review:**

no

**Detailed Comments:**

Some of the experimental details might be elaborated upon more; especially things like initialization of the network and what augmentations are used. These are relevant pieces of information on this more implementation focused paper.

**Final Rating Justification:**

I thank the reviewers for their comments and clarifications. Still, I would stay with my original comments and rating that the paper is decent but slightly below the acceptance threshold (relative to other papers). However, consider my confidence at "4", i.e. if other reviewers are for acceptance of the paper this should be weighed in as well.

**Justification Of The Preliminary Rating:**

The paper is interesting, using a combination of well known methods. However, the resulting system performs quite below the state of the art (of a difficult task with much competition, it should be noted) and the take-aways that certain choices in the model improve the performance, compared to certain comparable methods might be relevant, albeit slightly too weak for me to give this a weak accept.

**Paper Type:**

validation/application paper

**Questions To Address In The Rebuttal:**

As mentioned above, I'd like to see more details on the specifics of training the network.
Also, since the method does not quite outperform the best method, we might need more relevant 'take-aways' from the paper that make it valuable. The mentioned result that this upsampling strategy works better is interesting, although that alone is not quite enough, since this conclusion is also taken on this specific dataset in this specific setting.
I agree we shouldn't only look at SOTA results, but in absence of that the other qualities should maybe be highlighter more clearly.

**Special Issue:**

no

---

> ### Author Response · Authors · 2021-03-16
> **Feedback reviewer 3**
>
> Thanks for your feedback. Regarding your concerns:
>
> Weakness 1. As explained in the feedback of the other reviewers, we merely use 2.5D to save computation cost and time and to achieve comparability to the original DeepLesion baseline model. Our main contribution is the novel upsampling method which reduces the false positive rate, as shown in the results, compared to the other two methods, although it did not reach SOTA accuracies of 3D models.
>
> Weakness 2 and Detailed comment. Thanks so much for your comment. Some parts about the parameters are in fact not described clearly, and some parts are missing. We will add these parts in the new version and also list the parameters here:
>
> The CT images of DeepLesion are stored as unsigned 16 bit PNG files. We convert the values to attenuation coefficients. For normalization, we computed the 99th percentile of the attenuation coefficients once for the entire dataset and then divide each slice by that value. We decided against an instance-wise approach (like min-max scaling or zero-mean-unit-variance per slice) since the attenuation coefficients are constant for each type of tissue (assuming a fixed acquisition protocol) and thus give meaning to the gray values, which we hope the network to exploit. Each slice has 1 to 5 mm thickness in most cases, while some of the images are 0.625 or 2 mm. There is one key slice with 30 mm of extra slices in front of and behind the key slice for each lesion. However, only the key slice has the annotation data, including lesion types, coordinates of 2D bounding-boxes and RECIST diameters for the lesions. We have extracted only those CT images annotated with pulmonary nodules which resulted in 2,394 CT images. We split 1,916 CT images for training and 478 CT images for validation. The images were resized to 512x512 as input. For training, we adapted data augmentation where images are flipped horizontally and vertically, and are rotated in-plane at a random angle with a chance of 50% respectively to enrich the variability of the CT images. After augmenting the CT slices, the corresponding coordinates of bounding boxes are updated accordingly. The parameters of Adam were kept at default: learning rate=0.001, betas=(0.9, 0.999), eps=1e-08, weight_decay=0. The initialization of the layers’ parameters is also kept at default, i.e. Xavier uniform for kernels and zeros for biases.
>
> Questions to address in the rebuttal:
> We appreciate your comment pointing out the weaknesses of our paper, so we will add the training parameters. We are grateful for your comment about the trade-off between the SOTA and our specific contribution. In a new version, we will highlight that and make our main contribution of reducing the FP with the new upsampling method clearer.

---

### Official Review · ~Simeon_Emilov_Spasov1 · 2021-03-08

**Confidence:** 4
**Preliminary Rating:** 2
**Final Rating:** 2

**Summary:**

The goal of the paper is to produce a system for automatic pulmonary nodule detection. The primary methodological contribution is combining several existing approaches in one framework. More specifically, the authors use a Faster-RCNN detection network with a Dual Path Network backbone with skip-connections (U-Net style structure). Also, the authors use 2.5D image inputs produced by concatenating slices surrounding the lesion key slice.

**Strengths:**

1.	Results are reasonable although they do not outperform 3D CNNs.
2.	Authors provide an ablation study against Faster RCNN with a VGG16 backbone. They show multiscale features from the proposed decoder help for high quality RPN targets (outperform VGG16 backbone).
3.	The authors identify two strategies to implement upscaling (type 1 and 2) and experimentally validate type 2 gives better performance. Both ablation studies are necessary and important.


**Weaknesses:**

1.	A key issue with the paper is that the technical novelty is rather limited. The proposed framework seems like a compilation of known methods with a goal of creating a CAD system for pulmonary nodule detection with low computational requirements.  Also, the paper does not tackle a novel problem in itself. CAD systems for pulmonary nodule detection, although of high importance, are known and studied.
2.	One main motivation is keeping the computational effort of the model low but in fact the “computational effort” is never analyzed, e.g. parameter count, FLOPs, training/inference time, etc, compared to 2D/3D CNNs. Also, low parameter count/FLOPs and fast inference are important considerations for edge devices but why is this important in the context of pulmonary nodule detection when this can be done on a server?
3.	Given the goal of computational efficiency of the paper, the VGG16 baseline is 50% more efficient (“the baseline model requires only 50% of the time for computation”) compared to the DPN backbone. It seems like a trade-off between representational capacity vs computational efficiency. Isn’t the goal being computationally efficient *and* producing better results?


**Deanonymize Review:**

yes

**Final Rating Justification:**

I would like to thank the authors for their rebuttal. However, despite the additional explanations, I still consider the method very incremental. Comparing and analysing the computational effort between pure 2D vs 2.5D vs the 3D models would have been interesting for the rebuttal. I would not recommend acceptance for the paper.

**Justification Of The Preliminary Rating:**

I am lukewarm about this paper because it is a compilation of known methods with ok results. One of the main goals of the paper is to keep the computational effort of the method low but this is never evaluated.

**Paper Type:**

methodological development

**Special Issue:**

no

---

> ### Author Response · Authors · 2021-03-16
> **Feedback reviewer 2**
>
> Thank you for your feedback. Regarding your concerns:
>
> Weakness 1. We agree with you that our evaluation results did not outperform the SOTA compared with other related researches. The main contribution of this paper is to combine the dual path skip connections (including the new upsampling scheme) and using multi-scale feature maps in the classification sub-network to reduce the false positive rate in lung nodule detection in the CAD system. The results show that we can successfully reduce the FP rate with the proposed method with higher accuracy. If desired, we can change the motivation part in the introduction of the paper to illustrate our purpose more clearly and to reduce the confusion: We believe that this new method could help other works consider similar technical aspects in further research.
>
> Weakness 2 and 3. Our purpose of using 2.5D is trying to save computation cost for training and evaluating the models. 3D models inherently have higher requirements in terms of both time and computational resources compared to their “pure” 2D counterparts, i.e. only one slice as input. Of course, the 2.5D approach is not considered a methodological novelty but rather serves our purpose. Selecting the 2.5D approach was merely a design choice that we set at the beginning to achieve comparability with the VGG16 backbone model of the original DeepLesion architecture [1]. The DPN backbone definitely needs more computational power and time than the VGG16 baseline. But in fact, it is not our goal in this paper to optimize the computational efficiency by using 2.5D input but rather to incorporate some volumetric information without much additional computation needed. We will report the inference times and parameter counts of the different models in the paper, if still wished.
>
> [1] Yan K, Wang X, Lu L, Summers RM. DeepLesion: automated mining of large-scale lesion annotations and universal lesion detection with deep learning. J Med Imaging (Bellingham). 2018;5(3):036501. doi:10.1117/1.JMI.5.3.036501

---

### Official Review · AnonReviewer4 · 2021-03-09

**Confidence:** 4
**Preliminary Rating:** 1
**Recommendation:** Poster
**Final Rating:** 1

**Summary:**

This paper proposes a dual path network to extract multiscale feature maps to detect pulmonary nodules. Two types of upsampling strategies were proposed and compared in the decoder part. The proposed model were developed and validated with a public dataset and the performance were compared with existing methods as the benchamrk.

**Strengths:**

- The paper tackes clinically and technically important problem.
- The motivation of the study is plausible.
- Detailed information for the implementation such as experimental setup and network structure were given.

**Weaknesses:**

- The dataset used is not appropriate to demonstrate the effectiveness of the proposed method.
- The illustration of the proposed method is not satisfactory and confusing.
- The final performance of the proposed method is inferiror to existing methods.

**Deanonymize Review:**

no

**Detailed Comments:**

- Adding two more slices around the key slice is considered not enough to contain all the relevant information for target nodule.
- Training with 15 epochs would not be sufficient to reach the convegence of the model training.
- The choice of dataset(DeepLesion) is not seem to be approriate to evalute the lung nodule detection performance.

**Final Rating Justification:**

Even though the authors elaborated on the feedback and comments, the technical novelty and empricial result is still marginal and I keep the rating as 'strong reject'.

**Justification Of The Preliminary Rating:**

- The technical and clinical novelty of the proposed method is marginal.
- The dataset used for the study is not appropriate.
- The experiments are limited to support the motivation of using proposed method.

**Paper Type:**

methodological development

**Questions To Address In The Rebuttal:**

- As the authors argue that the 2.5D approach has avantange over better performing 3D models in terms of computational cost, it should be justified with assocated experiment.
- The dataset used for the development and validation of the proposed method is not appropriated. Consider using LUNA 16 dataset at least for the external performance validation.

**Special Issue:**

no

---

> ### Author Response · Authors · 2021-03-16
> **Feedback reviewer 4**
>
> Thanks for your feedback. Regarding your concerns:
>
> Weakness 1. We agree that the dataset is not the optimal choice. However, since DeepLesion does contain the data that is necessary to show the effectiveness of our method, i.e. lung nodules, we are positive that it is appropriate enough to be used for training and evaluating our models.
>
> Weakness 2. We are trying our best to describe our proposed method and will revise our paper. Of course, we would be very grateful if you can give more feedback on which part we should make it easier to follow.
>
> Weakness 3. The paper’s main contribution is to evaluate the proposed upsampling strategy, which shows the best performance compared with the other two methods. We also stated that our method is not SOTA concerning the dataset and GPU resources.
>
> Detailed Comment 1. In this paper, the purpose of using 2.5D is trying to save the computation cost for all the models. 3D models inherently have higher requirements in terms of both time and computational resources compared to their “pure” 2D counterparts, i.e. only one slice as input. We wanted to incorporate more volumetric information while not increasing computational cost much. Of course, the 2.5D approach is not considered a methodological novelty but rather serves our purpose.
>
> Detailed Comment 2. This depends on the convergence criterion. As you can see in Figure 4, the largest part of loss reduction has already taken place and we expect only small improvements afterwards. For this reason, and for the sake of reduced training time, we fixed the number of epochs to 15. Furthermore, the authors of the DeepLesion dataset report convergence of their detection model after only eight epochs [1].
>
> Detailed Comment 3. We agree that LUNA 16 would be one of the best choices for evaluating the nodule detection, while it is not the primary target that we would like to make it SOTA. Here, we mainly evaluate our proposed method to reduce the false positive rate by merging the skip connections in the Unet decoding path similarly to Dual Path Networks. The figure shows that we successfully reduce the FP with our proposed approach. We would like to apply this to LUNA16 later. However, in such a short time, we are afraid that we cannot make it.
>
> For the justification of the preliminary rating:
> 1. We agree with you that we did not achieve SOTA results in detecting lung nodules. However, our primary focus on reducing the FP has been successfully achieved. We believe this slight improvement will help in the future evaluations.
> 2. We understand the reviewer’s concerns. This dataset is not optimal, but good enough for the evaluation of our goals.
> 3. We are sorry that we did not make our contribution clear. The method we proposed to reduce the FP rate is expected to be independent of the dataset.
>
> [1] Yan K, Wang X, Lu L, Summers RM. DeepLesion: automated mining of large-scale lesion annotations and universal lesion detection with deep learning. J Med Imaging (Bellingham). 2018;5(3):036501. doi:10.1117/1.JMI.5.3.036501

---

### Meta-Review · Area_Chairs · 2021-03-29

**Recommendation:** Reject

**Metareview:**

The paper received unanimously negative reviewing comments from four independant reviewers, citing concerns about the limited novelty (mostly combination of known methods), possibly inappropriate choice of dataset, performances below SOTA ones, etc. The rebuttal does not sufficiently address the above concerns as most of the reviewers stick to the negative rating.

**Paper Type:**

methodological development

---

### Decision · Program_Chairs · 2021-03-31

Reject